# Association between intraoperative end-tidal carbon dioxide and postoperative organ dysfunction in major abdominal surgery: A cohort study

Li Dong[1,2,3]*, Chikashi Takeda[2], Tsukasa Kamitani[1], Miho Hamada[2], Akiko Hirotsu[2], Yosuke Yamamoto[1], Toshiyuki Mizota[2]

1 Department of Healthcare Epidemiology, Graduate School of Medicine and Public Health, Kyoto University, Kyoto, Japan, 2 Department of Anesthesia, Kyoto University Hospital, Kyoto, Japan, 3 Anesthesia Associates of Kobe, Kobe, Japan

* dongli@kuhp.kyoto-u.ac.jp

## Abstract

### Background

Data on the effects of intraoperative end-tidal carbon dioxide ($EtCO_2$) levels on postoperative organ dysfunction are limited. Thus, this study was designed to investigate the relationship between the intraoperative $EtCO_2$ level and postoperative organ dysfunction in patients who underwent major abdominal surgery under general anesthesia.

### Methods

We conducted a cohort study involving patients who underwent major abdominal surgery under general anesthesia at Kyoto University Hospital. We classified those with a mean $EtCO_2$ of less than 35 mmHg as low $EtCO_2$. The time effect was determined as the minutes when the $EtCO_2$ value was below 35 mmHg, whereas the cumulative effect was evaluated by measuring the area below the 35-mmHg threshold. The outcome was postoperative organ dysfunction, defined as a composite of at least one organ dysfunction among acute renal injury, circulatory dysfunction, respiratory dysfunction, coagulation dysfunction, and liver dysfunction within 7 days after surgery.

### Results

Of the 4,171 patients, 1,195 (28%) had low $EtCO_2$, and 1,428 (34%) had postoperative organ dysfunction. An association was found between low $EtCO_2$ and increased postoperative organ dysfunction (adjusted risk ratio, 1.11; 95% confidence interval [CI], 1.03–1.20; $p = 0.006$). Additionally, long-term exposure to $EtCO_2$ values of less than 35 mmHg ($\geq$224 min) was associated with postoperative organ dysfunction (adjusted risk ratio, 1.18; 95% CI, 1.06–1.32; $p = 0.003$) and low $EtCO_2$ severity (area under the threshold) (adjusted risk ratio, 1.13; 95% CI, 1.02–1.26; $p = 0.018$).

**Data Availability Statement:** The Institutional Review Board did not permit the sharing of our raw data (email: ethcom@kuhp.kyoto-u.ac.jp). In this

paper, there are ethical reasons for restricting the data. To minimize the risk of leaking personal information, the data extracted from medical records does not include information that directly identifies individuals such as name and date of birth. However, when submitting an ethical application to the ethics committee, we stated that we will not provide samples or information to other research institutions regarding the provision of samples or information used in the research, and regarding records related to the provision of samples or information.

**Funding:** This work was supported in part by the Japan Society for the Promotion of Science KAKENHI program (grant number: 20K09242; principal investigator: Toshiyuki Mizota) and the 2019 Kyoto University ISHIZUE Research Development Program (principal investigator: Toshiyuki Mizota). The funders had no role in study design, data collection and analysis, decision to publish, or preparation of the manuscript.

**Competing interests:** The authors have declared that no competing interests exist.

## Conclusions

Intraoperative low $EtCO_2$ of below 35 mmHg was associated with increased postoperative organ dysfunction.

## Introduction

Although high-risk surgeries account for only 12.5% of all surgical procedures, they account for more than 80% of surgery-related deaths [1]. Intraoperative organ hypoperfusion is a cause of poor outcomes and may lead to high postoperative mortality [2]. Therefore, markers that can be used to monitor intraoperative organ hypoperfusion and predict postoperative organ injury are essential to improve postoperative outcomes.

The International Standards for a Safe Practice of Anesthesia recommend monitoring end-tidal carbon dioxide ($EtCO_2$) using a capnograph during general anesthesia [3]. As $EtCO_2$ involves all four components of respiration and circulation (i.e., ventilation, diffusion, circulation, and metabolism), it provides an excellent picture of the respiratory and circulatory processes. Under conditions of constant ventilation, $EtCO_2$ can be used to monitor cardiac output and pulmonary blood flow [4, 5]. In fact, several studies have shown that $EtCO_2$ is useful in predicting the effectiveness of resuscitation [6] and outcomes in patients with cardiopulmonary arrest (CPA) [7, 8] and in predicting cardiac output when the patient is weaned from cardiopulmonary bypass (CPB) [9, 10]. Similarly, $EtCO_2$ in noncardiac surgery was associated with increased postoperative mortality [11] and prolonged postoperative length of hospital stay [11–13]. However, the association between $EtCO_2$ and postoperative organ dysfunction in patients undergoing general surgical treatments has not yet been fully evaluated.

Therefore, we investigated the association between intraoperative $EtCO_2$ and postoperative organ dysfunction in patients undergoing major abdominal surgery.

## Methods

### Study design, setting, and population

In this single-center cohort study, we used data from the IMProve Anesthesia Care and ouTcomes (Kyoto-IMPACT) database of Kyoto University Hospital. The Kyoto-IMPACT database aims to clarify the relationship between intraoperative respiratory and cardiovascular parameters and postoperative outcomes. We continuously selected patients who underwent surgery under the care of anesthesiologists at Kyoto University Hospital (1,121 beds). Several studies have been published using the Kyoto-IMPACT database [14, 15]. We included consecutive patients aged 18 years or older who underwent major abdominal surgery under general anesthesia at Kyoto University Hospital between March 2008 and December 2017. We included individuals who underwent abdominal surgery because major abdominal surgeries involve many cases, a long duration of surgery, and a high rate of postoperative organ dysfunction as outcomes. Major abdominal surgeries included laparoscopic or non-laparoscopic resections of the liver, colon, stomach, pancreas, and esophagus. The exclusion criteria were as follows: (1) patients with missing intraoperative $EtCO_2$ data; (2) second or subsequent surgery in patients who had undergone multiple surgeries; (3) patients undergoing urological surgery, such as urinary tract unblocking, nephrectomy, and renal transplantation; (4) patients undergoing renal replacement therapy for end-stage renal disease (estimated glomerular filtration rate of $< 15$ ml.min$^{-1}$.1.73 m$^{2-1}$); (5) patients with a preoperative platelet count of less than $100 \times 10^3$ cells.$\mu$l$^{-1}$; and (6) patients with a preoperative total bilirubin of more than or equal to 2.0 mg.dl$^{-1}$.

## Ethics

The Certified Review Board of Kyoto University (Yoshida-Konoe-cho, Sakyo-ku, Kyoto 606–8501, Japan, Chairperson Prof. Shinji Kosugi) approved the study protocol (approval number: R1272-3; January 23, 2020) and waived the requirement for informed consent because of the nature of this study.

## Data collection

We collected data from the Kyoto-IMPACT database using the anesthesia information management system and the electronic medical record system. $EtCO_2$ was measured continuously using a sidestream gas analyzer (GF-220R Multigas/Flow Unit, Nihon Kohden®, Japan), which was uploaded automatically to the anesthesia information management system every 60 s. We defined intraoperative $EtCO_2$ as the mean $EtCO_2$ level from skin incision to skin closure. $EtCO_2$ levels of less than 20 mmHg were removed as artifacts ($EtCO_2$ during aspiration or position change). Definitions of variables, including the minimum and maximum $EtCO_2$ values, can be found in S1 Table in S1 File. We collected data on patients' postoperative course (e.g., acute renal injury [AKI], circulatory dysfunction, respiratory dysfunction, coagulation dysfunction, and liver dysfunction within 7 days postoperatively) from all clinical data contained in the electronic medical records. Ventilator data can be found in S8 Table in S1 File.

## Exposure

To determine how the $EtCO_2$ level affects postoperative organ dysfunction, exposure was defined by calculating the dose, time, and cumulative effects of $EtCO_2$. Dose effects were assessed using the mean $EtCO_2$; patients were divided into two groups based on the cutoff level of 35 mmHg, widely used lower limit of normal $PaCO_2$ [16, 17]. We defined low $EtCO_2$ patients as those with a mean $EtCO_2$ of less than 35 mmHg, whereas we defined normal $EtCO_2$ patients as those with a mean $EtCO_2$ of more than or equal to 35 mmHg. The classification into one of these groups was used as the primary exposure for further analysis. Besides, we considered that the relationship between $EtCO_2$ and postoperative organ dysfunction may not be linear; thus, mean $EtCO_2$ values were classified into quartiles (i.e., <35, 35–37, 37–39, and $\geq$39 mmHg). To assess the effects of the duration and severity of low $EtCO_2$ exposure, time effects were determined as the minutes when $EtCO_2$ values were below 35 mmHg, and cumulative effects were assessed by measuring the area under the threshold of 35 mmHg for each patient. Additionally, we classified minutes and area under the $EtCO_2$ 35-mmHg threshold into quartiles, using the lowest quartile as the reference category.

## Outcomes

Referring to a previous study [18], the primary outcome was a composite of at least one organ dysfunction among AKI (postoperative serum creatinine [SCr] levels increased more than 0.3 mg.dl$^{-1}$ or 1.5 times more than preoperative SCr levels, defined by the Kidney Disease: Improving Global Outcome Acute Kidney Injury Work Group) [19], circulatory dysfunction (use of norepinephrine, epinephrine, and vasopressin and the administration of dopamine $\geq$5 μg.kg$^{-1}$.min$^{-1}$ and phenylephrine $\geq$50 μg.min$^{-1}$), respiratory dysfunction (the need for invasive ventilation by endotracheal intubation or tracheostomy beyond 24 h postoperatively; does not include continuous positive airway pressure or noninvasive ventilation or scheduled reintubation, such as extubation within 24 h after reoperation), coagulation dysfunction (platelet count of $< 100 \times 10^3$ cells.μl$^{-1}$, i.e., a Sequential Organ Failure Assessment [SOFA] score of $\geq$2 points in the coagulation component) [20], and liver dysfunction (total bilirubin of $\geq$ 2.0

mg.dl$^{-1}$, i.e., a SOFA score of $\geq 2$ points in the liver component) 7 days after surgery [20]. Secondary outcomes were individual components of the primary composite outcome.

## Statistical analyses

We planned to analyze the relationship between intraoperative EtCO$_2$ and postoperative organ dysfunction before data collection. Continuous variables were expressed as the median and interquartile range (IQR), and categorical variables were expressed as counts and percentages (%).

First, we performed multivariate Poisson regression with robust variance estimate [21] to calculate the risk ratios for low EtCO$_2$ (mean EtCO$_2$ of $< 35$ mmHg) and postoperative organ dysfunction, using the normal EtCO2 (mean EtCO$_2$ of $\geq 35$ mmHg) as the reference category. Additionally, the risk ratios for the mean EtCO$_2$ of the first quartile (mean EtCO$_2$ of $< 35$ mmHg), third quartile (mean EtCO$_2$ of 37–39 mmHg), and fourth quartile (mean EtCO$_2$ of $\geq 39$ mmHg) were calculated using the second quartile (mean EtCO$_2$ of 35–37 mmHg) as the reference category because it is considered normocapnia. Furthermore, to examine the time and cumulative effects, we evaluated how each quartile affected postoperative organ dysfunction, with the first quartile of minutes under an EtCO$_2$ of 35 mmHg and the area below the threshold EtCO$_2$ of 35 mmHg as reference categories.

To demonstrate the relationship between intraoperative EtCO$_2$ and postoperative organ dysfunction, we created four models using potential confounding factors that may be associated with the outcomes as follows. Model 1 included the covariates used for models 2, 3, and 4 and biologically and clinically essential data, including the body mass index, American Society of Anesthesiologists physical status (ASAPS), laparoscopic surgery, type of surgery, epidural anesthesia, and mean arterial pressure. In model 2, the covariates included in the AKI risk index were adjusted for age equal to or older than 56 years, male sex, emergency surgery, diabetes mellitus, active congestive heart failure, ascites, hypertension, and renal insufficiency [22]. In model 3, the covariates included in the Revised Cardiac Risk Index (RCRI) were adjusted for emergency surgery, surgery duration longer than 4 h, ischemic heart disease, congestive heart failure, cerebrovascular disease, diabetes, and perioperative SCr of more than 2.0 mg.dl$^{-1}$ [23]. In model 4, the covariates included in the postoperative respiratory failure risk index (RFRI) were adjusted for age, emergency surgery, albumin level of less than 30 g.l$^{-1}$, blood urea nitrogen level of more than or equal to 30 mg.dl$^{-1}$, and chronic obstructive pulmonary disease (COPD) [24], except for partially or fully dependent status because of missing data. Additionally, we adjusted for the aforementioned multivariate regression models to investigate whether the dose, time, or cumulative effects of EtCO$_2$ affected the secondary outcomes.

The categories of EtCO$_2$ were treated as continuous variables, and the mean, minutes, and area under the threshold of categories were substituted into the multivariate Poisson regression model, with the median of each group as independent variables. A test of the linear trend was performed between the categories of EtCO$_2$ and postoperative organ dysfunction, adjusted using the aforementioned model 1 (P for trend).

We performed a sensitivity analysis to assess the robustness of our findings. First, to assess the plausibility of the primary analysis, we performed a multivariate analysis using the sensitivity model described above as model 1: (i) patients for whom arterial gases were measured during surgery (ii) patients for whom minute ventilation data were available. For patients with intraoperative arterial gas measurements, the arterial partial pressure of carbon dioxide (PaCO$_2$)-EtCO$_2$ gradient was added to the covariates in model 1, defined as model 5, and the intraoperative maximum lactate value was added, defined as model 6, to investigate the

relationship between intraoperative EtCO2 and postoperative organ dysfunction. Second, for patients for whom minute ventilation data were available, median minute ventilation was added to the covariates in model 1, defined as model 7 and multivariate analysis was performed. Finally we used the same multivariate regression models in subgroup analysis of laparoscopic versus open surgery. We calculated the crude risk ratio for postoperative organ dysfunction in each subgroup and tested the interaction between subgroups and EtCO2.

To maximize statistical power, all eligible patients in the Kyoto-IMPACT database were included in the analyses. To determine the statistical power, we predicted 4,500 eligible surgeries in our database in the 9 years, a risk ratio of 1.5 with postoperative organ dysfunction of 30% [18] and low $EtCO_2$ proportion of 50% [25], resulting in an estimated power of 100%. We conducted complete case analysis because the percentage of missing data was 0.12%. All statistical tests were two-tailed. In all statistical analyses, Stata/SE 15.1 (StataCorp LLC, College Station, TX, USA) was used.

## Results

### Baseline patient characteristics

Among the 4,781 patients who underwent major abdominal surgeries between 2008 and 2017, 4,772 met the inclusion criteria and were included in the analyses (4,171 were complete cases) (Fig 1). Low $EtCO_2$ (defined as a mean $EtCO_2$ of < 35 mmHg) occurred in 28% of the patients included. Table 1 displays the characteristics of the study participants. The median $EtCO_2$ level

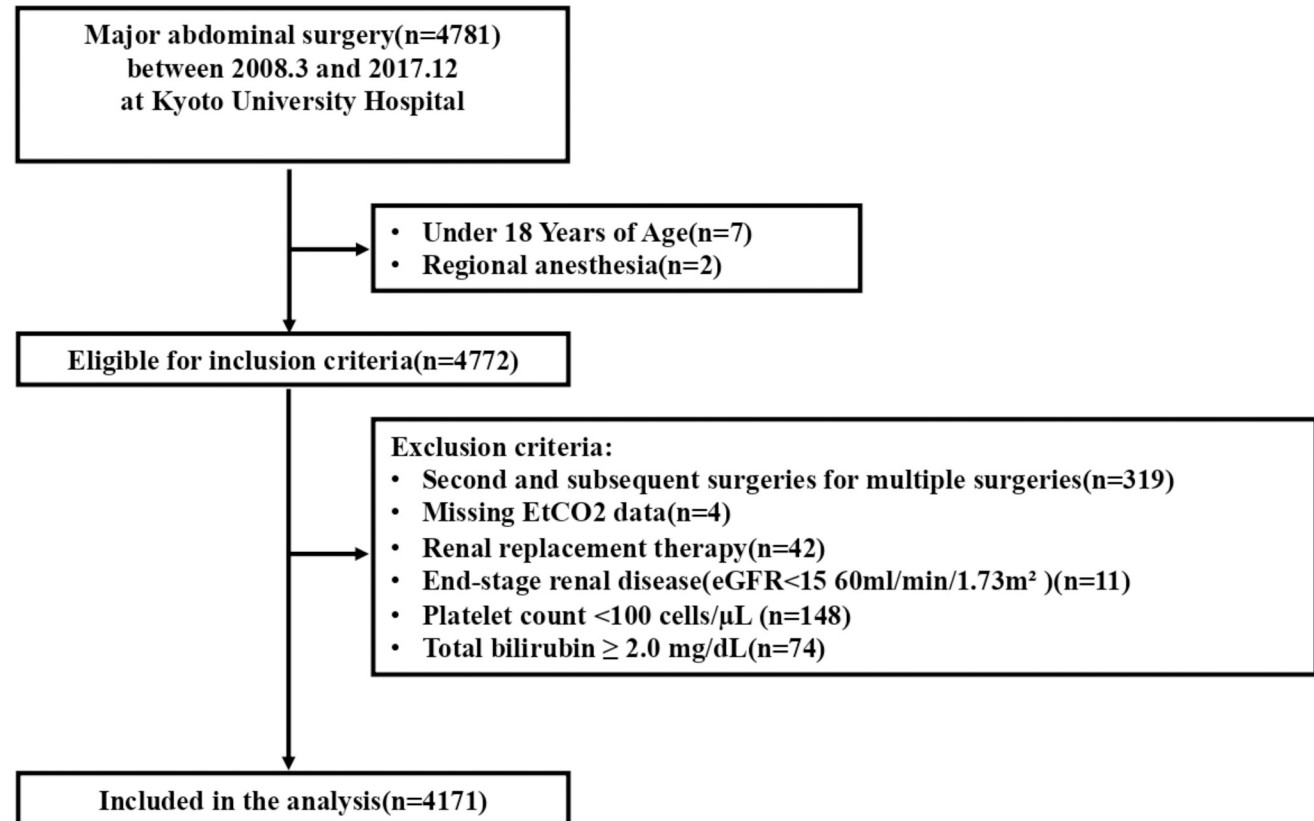

**Fig 1. Flowchart of this study.** We included consecutive patients aged 18 years or older who underwent major abdominal surgery under general anesthesia at Kyoto University Hospital between 2012 and 2017. Then, we extracted the cases that met our eligibility criteria and analyzed them as complete cases.

**Table 1. Patients' characteristics (n = 4,171).**

| Characteristics | All patients (n = 4,171) | Low EtCO$_2$ (n = 1,195) | Normal EtCO$_2$ (n = 2,976) | P value |
|---|---|---|---|---|
| Age, years | 66 (56–74) | 68 (57–75) | 65 (56–73) | <0.001 |
| Male gender | 2551 (61.16%) | 702 (58.74%) | 1849 (62.13%) | 0.051 |
| ASAPS | | | | <0.001 |
| I | 1181 (28.31%) | 300 (25.10%) | 881 (29.60%) | |
| II | 2727 (65.38%) | 792 (66.28%) | 1935 (65.02%) | |
| III | 256 (6.14%) | 99 (8.28%) | 157 (5.28%) | |
| IV | 7 (0.17%) | 4 (0.33%) | 3 (0.10%) | |
| BMI, kg.m$^{-2}$ | 22 (20–24) | 22 (20–24) | 22 (19–24) | 0.375 |
| COPD | 391 (9.37%) | 130 (10.88%) | 261 (8.77%) | 0.041 |
| Albumin level < 30 g.l$^{-1}$ | 206 (4.94%) | 70 (5.86%) | 136 (4.57%) | 0.059 |
| Type of surgery | | | | <0.001 |
| Colorectal | 1297 (31.10%) | 350 (29.29%) | 947 (31.82%) | |
| Liver | 1202 (28.82%) | 434 (36.32%) | 768 (25.81%) | |
| Gastric | 779 (18.68%) | 140 (11.72%) | 639 (21.47%) | |
| Pancreatic | 615 (14.74%) | 214 (17.91%) | 401 (13.47%) | |
| Esophageal | 253 (6.07%) | 50 (4.18%) | 203 (6.82%) | |
| Complex | 25 (0.60%) | 7 (0.59%) | 18 (0.60%) | |
| Laparoscopic surgery | 2529 (60.63%) | 552 (46.19%) | 1977 (66.43%) | <0.001 |
| Emergency surgery | 36 (0.86%) | 15 (1.26%) | 21 (0.71%) | 0.256 |
| Epidural anesthesia | 1479 (35.46%) | 503 (42.09%) | 976 (32.80%) | <0.001 |
| Duration of surgery, hours | 5.88 (4.31–7.88) | 6 (4.45–7.98) | 5.85 (4.28–7.81) | 0.128 |
| MAP, mmHg | 74 (69–80) | 73 (68–79) | 74 (69–80) | <0.001 |
| Blood loss, ml | 125 (24–460) | 260 (50–669) | 95 (20–365) | <0.001 |
| Transfusion volume, ml | 0 (0–0) | 0 (0–0) | 0 (0–0) | <0.001 |
| Infusion volume, ml | 3070 (2100–4330) | 3430 (2340–4700) | 2900 (2050–4200) | <0.001 |
| Charlson Comorbidity Index | 4 (2–5) | 4 (2–6) | 3 (2–5) | 0.067 |
| Mean RR, rpm | 11 (10–12) | 10 (10–12) | 11 (10–12) | 0.072 |
| Mean FiO$_2$, % | 41 (35–45) | 41 (36–45) | 41 (35–45) | 0.303 |
| Mean BIS | 44 (40–50) | 44 (39–49) | 44 (40–50) | 0.082 |
| Mean EtCO$_2$ | 36 (34–39) | 33 (31–34) | 38 (36–40) | <0.001 |
| Minimum EtCO$_2$ | 30 (28–33) | 27 (25–29) | 32 (29–33) | <0.001 |
| Maximum EtCO$_2$ | 43 (40–46) | 38 (36–40) | 44 (42–48) | <0.001 |

Values are given as the median (interquartile range) or count (%).

Abbreviations: ASAPS, American Society of Anesthesiologists Physical Status; BMI, body mass index; COPD, chronic obstructive pulmonary disease; MAP, mean arterial pressure; RR, respiratory rate; FiO2, fraction of inspiratory oxygen; BIS, bispectral index; EtCO$_2$, end-tidal carbon dioxide.

was 36 mmHg (IQR, 34–39 mmHg) for the entire population, 33 mmHg (IQR, 31–34 mmHg) for patients with low EtCO$_2$, and 38 mmHg (IQR, 36–40 mmHg) for patients with normal EtCO$_2$.

## Association between low EtCO$_2$ and postoperative organ dysfunction

Table 2 shows the main results of this study. Postoperative organ dysfunction was observed in 41.67% (498 of 1,195 patients) in the low EtCO$_2$ group compared with 31.25% (930 of 2,976 patients) in the normal EtCO$_2$ group. The adjusted risk ratio by multivariate Poisson regression analysis for the low EtCO$_2$ group (mean EtCO$_2$ of < 35 mmHg) suggested an association between low EtCO$_2$ and postoperative organ dysfunction (model 1 adjusted risk ratio, 1.11; 95% confidence interval [CI], 1.03–1.20; $p = 0.006$).

**Table 2. Multivariable analysis of the relationship between $EtCO_2$ and organ dysfunction.**

| | Organ dysfunction (%) | Crude RR (95% CI) | P-value | Adjusted RR (95% CI) Model 1‡ | P-value | P for trend |
|---|---|---|---|---|---|---|
| **Mean $EtCO_2$** | | | | | | |
| Low $EtCO_2$ | 498/1195 (41.67%) | 1.33 (1.22–1.45) | <0.001 | 1.11 (1.03–1.20) | 0.006 | |
| Normal $EtCO_2$ | 930/2976 (31.25%) | 1 | – | 1 | – | |
| **Mean $EtCO_2$** | | | | | | 0.002 |
| <35 mmHg | 498/1195 (41.67%) | 1.13 (1.02–1.25) | 0.019 | 1.08 (0.99–1.19) | 0.071 | |
| 35–37 mmHg | 369/1004 (36.75%) | 1 | – | 1 | – | |
| 37–39 mmHg | 294/864 (34.03%) | 0.92 (0.81–1.04) | 0.221 | 1.01 (0.90–1.12) | 0.830 | |
| ≥39 mmHg | 267/1108 (24.10%) | 0.65 (0.57–0.74) | <0.001 | 0.90 (0.80–1.01) | 0.097 | |
| **Minutes below $EtCO_2$ 35 mmHg** | | | | | | 0.003 |
| Quartile value 1 (0–20 min) | 275/1,059 (25.97%) | 1 | – | 1 | – | |
| Quartile value 2 (21–93 min) | 297/1,063 (27.94%) | 1.07 (0.93–1.23) | 0.306 | 1.09 (0.96–1.23) | 0.166 | |
| Quartile value 3 (94–223 min) | 331/1,038 (31.89%) | 1.22 (1.07–1.40) | 0.003 | 1.02 (0.90–1.15) | 0.746 | |
| Quartile value 4 (224–1,069 min) | 525/1,011 (51.93%) | 1.99 (1.77–2.24) | <0.001 | 1.18 (1.06–1.32) | 0.003 | |
| **Area under the threshold of $EtCO_2$ 35 mmHg** | | | | | | 0.001 |
| Quartile value 1 (0–13 mm.Hg.min⁻¹) | 306/1,048 (29.20%) | 1 | – | 1 | – | |
| Quartile value 2 (14–102 mm.Hg.min⁻¹) | 284/1,062 (26.74%) | 0.91 (0.79–1.05) | 0.209 | 0.94 (0.83–1.06) | 0.369 | |
| Quartile value 3 (103–405 mm.Hg.min⁻¹) | 370/1,044 (35.44%) | 1.21 (1.07–1.37) | 0.002 | 1.05 (0.94–1.17) | 0.351 | |
| Quartile value 4 (406–6574 mm.Hg.min⁻¹) | 468/1,017 (46.02%) | 1.57 (1.40–1.76) | <0.001 | 1.13 (1.02–1.26) | 0.018 | |

Model 1‡: BMI, ASAPS 3 and above, laparoscopic surgery, type of surgery, epidural anesthesia, MAP, age, gender, emergency surgery, diabetes mellitus, congestive heart failure, ascites, hypertension, chronic kidney disease, surgery duration longer than 4 h, ischemic heart disease, cerebrovascular disease, albumin level of less than 30 g.l⁻¹, and COPD.

Abbreviations: $EtCO_2$, end-tidal carbon dioxide; CI, confidence interval; RR, risk ratio; BMI, body mass index; ASAPS, American Society of Anesthesiologists Physical Status; MAP, mean arterial pressure; COPD, chronic obstructive pulmonary disease.

For further analysis, $EtCO_2$ was divided into quartiles, and the second quartile (mean $EtCO_2$ of 35–37 mmHg) was used as the reference and as the definition of low $EtCO_2$ (the lowest quartile of the mean $EtCO_2$ values [mean $EtCO_2$ of < 35 mmHg]). Postoperative organ dysfunction decreased gradually from the first to the fourth quartiles (first quartile, 41.67%; second quartile, 36.75%; third quartile, 34.03%; and fourth quartile, 24.10%). The multivariate adjusted risk ratios were 1.08 (95% CI, 0.99–1.19) for the first quartile, 1.01 (95% CI, 0.90–1.12) for the third quartile, and 0.90 (95% CI, 0.80–1.01) for the fourth quartile, with the second quartile being used as the reference (*p* for trend = 0.002) (Table 2).

Regarding the time effect of $EtCO_2$, compared with short-term exposure (first quartile of exposure time to $EtCO_2$ of < 35 mmHg, 0–20 min), long-term exposure to $EtCO_2$ levels of less than 35 mmHg (the fourth quartile of exposure time to $EtCO_2$ of < 35 mmHg, 224–1,069 min) was associated with increased postoperative organ dysfunction (model 1 adjusted risk ratio, 1.18; 95% CI, 1.06–1.32; *p* = 0.003). Finally, for the cumulative effect of $EtCO_2$, the fourth quartile of the area below the $EtCO_2$ threshold of 35 mmHg (406–6,574 mmHg) was associated with increased organ dysfunction compared with the first quartile (model 1 adjusted risk ratio, 1.13; 95% CI, 1.02–1.26; *p* = 0.018).

## Association between low $EtCO_2$ and secondary outcomes

Table 3 shows the relationship between low $EtCO_2$ and postoperative AKI (model 1 adjusted risk ratio, 1.04; 95% CI, 0.82–1.32; *p* = 0.712), postoperative circulatory dysfunction (model 1 adjusted odds ratio, 1.44; 95% CI, 0.79–2.60; *p* = 0.229), postoperative respiratory dysfunction (model 1 adjusted odds ratio, 2.19; 95% CI, 1.09–4.39; *p* = 0.026), postoperative coagulation

**Table 3. Multivariable analysis of the relationship between $EtCO_2$ and secondary outcomes.**

| AKI | AKI (%) | Crude RR (95% CI) | P value | Adjusted RR (95% CI) Model 1‡ | *P*-value | Adjusted RR (95% CI) Model 2§ | *P*-value |
|---|---|---|---|---|---|---|---|
| Low $EtCO_2$ | 98/1,195 (8.20%) | 1.27 (1.00–1.60) | 0.044 | 1.04 (0.82–1.32) | 0.712 | 1.04 (0.85–1.28) | 0.667 |
| Normal $EtCO_2$ | 192/2,976 (6.45%) | 1 | – | 1 | – | 1 | – |
| **Circulatory dysfunction** | **Circulatory dysfunction (%)** | | | | | **Model 3§** | |
| Low $EtCO_2$ | 23/1,195 (1.92%) | 1.73 (1.02–2.94) | 0.041 | 1.44 (0.79–2.60) | 0.229 | 1.47 (0.82–2.62) | 0.187 |
| Normal $EtCO_2$ | 33/2,976 (1.11%) | 1 | – | 1 | – | 1 | – |
| **Respiratory dysfunction** | **Respiratory dysfunction (%)** | | | | | **Model 4§** | |
| Low $EtCO_2$ | 16/1,195 (1.34%) | 2.49 (1.24–4.96) | 0.010 | 2.19 (1.09–4.39) | 0.026 | 2.18 (1.11–4.31) | 0.024 |
| Normal $EtCO_2$ | 16/2,976 (0.54%) | 1 | – | 1 | – | 1 | – |
| **Coagulation dysfunction** | **Coagulation dysfunction (%)** | | | | | | |
| Low $EtCO_2$ | 212/1,195 (17.74%) | 1.63 (1.39–1.91) | <0.001 | 1.12 (0.96–1.31) | 0.136 | | |
| Normal $EtCO_2$ | 323/2,976 (10.85%) | 1 | – | 1 | – | | |
| **Liver dysfunction** | **Liver dysfunction (%)** | | | | | | |
| Low $EtCO_2$ | 339/1,195 (28.37%) | 1.31 (1.17–1.47) | <0.001 | 1.10 (0.99–1.21) | 0.054 | | |
| Normal $EtCO_2$ | 641/2,976 (21.54%) | 1 | – | 1 | – | | |

Model 1‡: BMI, ASAPS 3 and above, laparoscopic surgery, type of surgery, epidural anesthesia, MAP, age, gender, emergency surgery, diabetes mellitus, congestive heart failure, ascites, hypertension, chronic kidney disease, surgery duration longer than 4 h, ischemic heart disease, cerebrovascular disease, albumin level of less than 30 g.l⁻¹, and COPD.

Model 2§: age more than 56 years, gender, congestive heart failure, ascites, hypertension, chronic kidney disease, diabetes mellitus, and emergency surgery.

Model 3§: ischemic heart disease, congestive heart failure, cerebrovascular disease, diabetes mellitus, chronic kidney disease, surgery duration longer than 4 hours, emergency surgery, and type of surgery.

Model 4§: age, emergency surgery, albumin level <30 g.l⁻¹, blood urea nitrogen level ≥30 mg.dl⁻¹, and COPD.

Abbreviations: AKI, acute kidney injury; $EtCO_2$, end-tidal carbon dioxide; CI, confidence interval; RR, risk ratio; BMI, body mass index; ASAPS, American Society of Anesthesiologists Physical Status; MAP, mean arterial pressure; COPD, chronic obstructive pulmonary disease.

dysfunction (model 1 adjusted risk ratio, 1.63; 95% CI, 1.39–1.91; $p < 0.001$), and postoperative liver dysfunction (model 1 adjusted risk ratio, 1.10; 95% CI, 0.99–1.21; $p = 0.054$). S2–S6 Tables in S1 File show the relationships between the quartiles of $EtCO_2$ and secondary outcomes and the time and cumulative effects of $EtCO_2$ on secondary outcomes.

## Sensitivity analysis

In the sensitivity analysis, the association between low $EtCO_2$ and postoperative organ dysfunction was observed even when the study was restricted to patients who had arterial gas measurements during surgery and for whom minute ventilation data were available. Even after adjusting for the $PaCO_2$-$EtCO_2$ gradient, intraoperative maximal lactate or minute ventilation, low $EtCO_2$ was associated with increased postoperative organ dysfunction (S7 Table in S1 File). In the subgroup analysis of laparoscopic versus open surgery, the point estimates of the risk ratio for organ dysfunction were exceeded than 1 (S9 Table in S1 File).

## Discussion

### Overview of the results

In this cohort study, postoperative organ dysfunction occurred in 41.67% of the patients in the low $EtCO_2$ group and in 31.25% of the patients in the normal $EtCO_2$ group. We found that

intraoperative $EtCO_2$ was associated with a 1.11-fold increase in postoperative organ dysfunction.

## Comparison with previous studies

Studies on $EtCO_2$ have focused primarily on patients with CPA and assessed the usefulness of $EtCO_2$ as a valuable tool that can assess the effectiveness of cardiopulmonary resuscitation (CPR) [6–9] and predict the outcome. This is because $EtCO_2$ correlates well with cardiac output, and no other appropriate noninvasive methods exist to measure this important variable during CPR. Furthermore, although $EtCO_2$ has been reported to correlate with cardiac output during withdrawal from CPB [10] and can predict volume responsiveness [26], studies have not evaluated the relationship between $EtCO_2$ and clinical outcomes, and the mechanism of this association is unclear [10]. Some studies have reported that $EtCO_2$ is associated with increased postoperative mortality [11] and prolonged postoperative length of hospital stay [12, 13], but the cause of death has not been evaluated, and the mechanism is unknown. Furthermore, no studies have focused on the association between intraoperative $EtCO_2$ and postoperative organ dysfunction in patients undergoing general surgery. In this large cohort study, we could demonstrate a dose-dependent association between intraoperative $EtCO_2$ and postoperative organ dysfunction using multivariate Poisson regression analysis in patients undergoing high-risk major abdominal surgeries.

## Mechanism

As for the mechanism of the association between intraoperative $EtCO_2$ and postoperative organ dysfunction, we interpreted that low cardiac output is associated with hypotension and low $EtCO_2$, resulting in intraoperative hypoperfusion and postoperative organ dysfunction, when ventilation is constant during surgery. Additionally, even if blood pressure is stabilized by increasing peripheral vascular resistance to compensate for low cardiac output, low $EtCO_2$ because of low cardiac output is associated with postoperative organ dysfunction, regardless of blood pressure. As an alternative to markers of cardiac output, such as cardiac output from pulmonary artery catheters or noninvasive cardiac output monitors, $EtCO_2$ levels may provide an objective assessment of cardiac output and organ perfusion status.

## Clinical implication

As $EtCO_2$ is monitored regularly in patients undergoing general anesthesia, it may serve as an early indicator of intraoperative low cardiac output, organ hypoperfusion, and postoperative organ dysfunction. This study showed that even after adjusting for intraoperative blood pressure, $EtCO_2$ was associated with postoperative organ dysfunction. Intraoperative $EtCO_2$ values, along with other vital signs, can comprehensively assess a patient's cardiac output and organ perfusion status in terms of ventilation, diffusion, circulation, and metabolism. As a practical clinical implication, $EtCO_2$ may help make clinical decisions to optimize organ perfusion, such as whether vasoactive drugs should be added because of decreased afterload, inotropes should be added because of decreased preload, or fluids or blood transfusions should be given because of bleeding.

## Strengths

This study has several strengths. First, this study investigated not only the dose effect of the mean $EtCO_2$ level of less than 35 mmHg but also the effect of prolonged exposure to $EtCO_2$ levels of less than 35 mmHg ($\geq$224 min) and the severity of low $EtCO_2$ exposure (area below

the threshold). Second, we adjusted not only for various factors included in the existing AKI risk index, RCRI, and RFRI but also for various confounding factors, such as laparoscopic surgery, type of surgery, intraoperative blood pressure, and ASAPS in four models. Third, there were few missing data, and 99.9% of the cases were complete.

## Limitations

This study has several limitations. First, the main analysis of this study did not consider the $PaCO_2$–$EtCO_2$ gap to calibrate $EtCO_2$ concentrations using $PaCO_2$ levels. $PaCO_2$ is usually 2–5 mmHg higher than $EtCO_2$ in healthy populations. Thus, this study underestimated the effect of low $EtCO_2$ and overestimated the effect of hypercapnia. However, when restricted to patients with intraoperative arterial gas measurements, low $EtCO_2$ was associated with increased postoperative organ dysfunction, even when $PaCO_2$-$EtCO_2$ gradient was adjusted. Second, as an unknown confounder, we did not know the potential reasons for anesthesiologists to target a specific $EtCO_2$ level. Anesthesiologists' interpretation of $EtCO_2$ may influence their decisions regarding anesthesia management. Third, unmeasured confounding factors, such as smoking history, intraoperative medications, and intraoperative ventilation parameters, may have influenced the association between intraoperative $EtCO_2$ and postoperative organ dysfunction. For example, hyperventilation is often indicated for patients with intraoperative hyperkalemia, which may lower the level of $EtCO_2$, thus probably overestimating the effects of low $EtCO_2$. Alternatively, it may underestimate the effects of low $EtCO_2$ because of the intravenous administration of sodium bicarbonate in patients with acidosis, thus probably raising the level of $EtCO_2$. Fourth, as this was an observation-based study, it does not show causality and could not confirm whether intraoperative management targeting an intraoperative $EtCO_2$ level of 35 mmHg or higher reduces postoperative organ dysfunction.

## Conclusion

In conclusion, in patients undergoing major abdominal surgery, intraoperative low $EtCO_2$ levels of less than 35 mmHg were associated with increased postoperative organ dysfunction, suggesting that intraoperative $EtCO_2$ is a predictor of postoperative organ dysfunction.

## Supporting information

**S1 Checklist. STROBE statement—Checklist of items that should be included in reports of observational studies.**
(DOC)

**S1 File.**
(DOCX)

## Acknowledgments

**Assistance with the study:** We are grateful to Mr. Yoshihiro Kinoshita, Ms. Tomoko Hosoya, and Mr. Yohei Taniguchi (Medical Information Systems Section, Management Division, Kyoto University Hospital, Kyoto, Japan) for their assistance in data collection for this study.

## Author Contributions

**Conceptualization:** Li Dong, Chikashi Takeda, Tsukasa Kamitani, Miho Hamada, Akiko Hirotsu, Yosuke Yamamoto, Toshiyuki Mizota.

**Data curation:** Li Dong, Yosuke Yamamoto, Toshiyuki Mizota.

**Formal analysis:** Li Dong, Chikashi Takeda, Miho Hamada, Akiko Hirotsu, Yosuke Yamamoto, Toshiyuki Mizota.

**Funding acquisition:** Toshiyuki Mizota.

**Investigation:** Li Dong, Chikashi Takeda, Tsukasa Kamitani, Miho Hamada, Yosuke Yamamoto, Toshiyuki Mizota.

**Methodology:** Li Dong, Chikashi Takeda, Tsukasa Kamitani, Yosuke Yamamoto, Toshiyuki Mizota.

**Project administration:** Li Dong, Yosuke Yamamoto, Toshiyuki Mizota.

**Resources:** Yosuke Yamamoto, Toshiyuki Mizota.

**Software:** Li Dong, Yosuke Yamamoto, Toshiyuki Mizota.

**Supervision:** Li Dong, Toshiyuki Mizota.

**Validation:** Li Dong, Chikashi Takeda, Tsukasa Kamitani, Miho Hamada, Akiko Hirotsu, Yosuke Yamamoto, Toshiyuki Mizota.

**Visualization:** Li Dong, Chikashi Takeda, Tsukasa Kamitani, Miho Hamada, Akiko Hirotsu, Yosuke Yamamoto, Toshiyuki Mizota.

**Writing – original draft:** Li Dong.

**Writing – review & editing:** Li Dong, Chikashi Takeda, Tsukasa Kamitani, Miho Hamada, Akiko Hirotsu, Yosuke Yamamoto, Toshiyuki Mizota.

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
