## [Decision Letter · Decision Letter 0]

1 Sep 2022

PONE-D-22-12191Association between Intraoperative End-Tidal Carbon Dioxide and Postoperative Organ Dysfunction in Major Abdominal Surgery: A Retrospective Cohort StudyPLOS ONE

Dear Dr. Dong,

Thank you for submitting your manuscript to PLOS ONE. After careful consideration, we feel that it has merit but does not fully meet PLOS ONE’s publication criteria as it currently stands. Therefore, we invite you to submit a revised version of the manuscript that addresses the points raised during the review process. I appologize for the delay in reviewing this manuscript, but it proved extraordinary difficult to recruit knowledgeable reviewers. Please refer to the reviewers comments for guidance for the manuscript revision.

We look forward to receiving your revised manuscript.

Kind regards,

Jörn Karhausen

Academic Editor

PLOS ONE

Journal Requirements:

Additional Editor Comments (if provided):

This very interesting study of Dong et al. explores the relationship of EtCO2 and organ function after major abdominal surgery based on a concept where EtCO2 serves as a summary indicator of organ perfusion and on several reports mostly from resuscitation setting where low EtCO2`s are linked to poor outcomes. The intraoperative period offers a much more defined setting vs. cardiopulmonary arrest including continuous EtCO2 values, more reliable ventilation settings and the capture of ventilation variables, therefore this large, single center, retrospective study likely yields a much more reliable statement on the utililty of EtCO2 as a predictor of outcomes. Importantly, since EtCO2 measurements are standard of care, this clinical variable is readily available in the perioperative period.

However, there are two major issues remaining in this study:

Due to the potential of CO2 retention in laparoscopic procedures, inclusion of these procedures next to open procedures may be problematic. The impact or lack thereof of these studies should be examined specifically i.e. using sensitivity analysis.

Since the authors hypothesize that EtCO2 is mainly an indicator of perfusion a major limitation of this study is that the relation of EtCO2 and intraoperative blood pressure measurements was not analyzed in more detail. I.E. the claim is made in the discussion that hypotension and low EtCO2, result in intraoperative hypoperfusion and postoperative organ dysfunction but only a mean MAP is shown in Table 1 without further analysis. Please include a more formal examination of EtCO2 to MAP e.g. using mean MAP and time below a threshold MAP in parallel to the model used for EtCO2. Along the same lines intraoperative vasopressor use and lactate levels should be included in the analysis.

Minor issues:

• Please include p values for comparisons of groups in Table 1

Reviewers' comments:

Reviewer's Responses to Questions

**Comments to the Author**

1. Is the manuscript technically sound, and do the data support the conclusions?

Reviewer #1: Partly

2. Has the statistical analysis been performed appropriately and rigorously? 

Reviewer #1: Yes

3. Have the authors made all data underlying the findings in their manuscript fully available?

Reviewer #1: No

4. Is the manuscript presented in an intelligible fashion and written in standard English?

Reviewer #1: Yes

5. Review Comments to the Author

Reviewer #1: The authors reported about the results of cohort study based on an existing data set to investigate the association between intraoperative end-tidal carbon dioxide and postoperative organ dysfunction. They concluded that intraoperative low EtCO2 of below 35 mmHg was associated with increased postoperative organ dysfunction.

To my understanding some aspects - in particular with respect to the results should be elaborated.

First, the term "retrospective" connected to the study design should be avoided all through the paper according to the STROBE guideline for cohort studies.

Please organize the paper according to the STROBE guideline for cohort studies and provide the checklist.

I have some problems with understanding the "mean EtCO2" and its dichotomization to below and above 35 mmHG. Exclusion of EtCO2 < 20 mmHG (as artefact) has major impact on the number of cases with low EtCO2, as the mean EtCO2 is increased. The number of measurements also impacts the rate. As mean EtCO2 "smoothers" the effect I ask myself, if this is the clinical important effect. And an argument for the cutoff might be important. Note that dichotomization is in general problematic, for instance because of measurement errors.

Second in the definition of the primary endpoint variable the time to event is not reflected. This might be worth to analyze. Moreover just counting events assumes the similar importance of the events, which is rarely true in practice. So why not study the events separately, with time to event models.

6. PLOS authors have the option to publish the peer review history of their article (what does this mean?). If published, this will include your full peer review and any attached files.

Reviewer #1: No

---

## [Author Response · Author response to Decision Letter 0]

29 Oct 2022

Ref: PONE-D-22-12191

Academic Editor

PLOS ONE

Dear Prof. Jörn Karhausen:

We would like to thank all of you for providing us with valuable comments on our manuscript entitled “Association between Intraoperative End-Tidal Carbon Dioxide and Postoperative Organ Dysfunction in Major Abdominal Surgery: A Retrospective Cohort Study” We have responded to all the reviewers' comments in"Response to Reviewers_R1". Our itemized responses are listed below. We hope this revision process has remedied all the concerns related to the previous version of the manuscript. Changes or corrections of the original manuscript have been highlighted in red in the revised manuscript. The revised manuscript has also been checked and revised by a native English-speaking medical-editing professional.

We hope your response to our revised manuscript will be positive. Thank you once again for reviewing it.

Sincerely,

Correspondence Author: 

Li Dong, M.D.P.D.

Department of Anaesthesia, Kyoto University Hospital, 54 Shogoin-Kawahara-cho, Kyoto 606-8507, JAPAN

Tel: +81-75-751-3433

E-mail: dongli@kuhp.kyoto-u.ac.jp

Response to Additional Editor 

We wish to express our appreciation to the referee for the insightful comments that have helped us improve our manuscript significantly. We agree with all the comments and have incorporated them into the revised version of our manuscript (R1).

Additional Editor Comments (if provided):

This very interesting study of Dong et al. explores the relationship of EtCO2 and organ function after major abdominal surgery based on a concept where EtCO2 serves as a summary indicator of organ perfusion and on several reports mostly from resuscitation setting where low EtCO2`s are linked to poor outcomes. The intraoperative period offers a much more defined setting vs. cardiopulmonary arrest including continuous EtCO2 values, more reliable ventilation settings and the capture of ventilation variables, therefore this large, single center, retrospective study likely yields a much more reliable statement on the utililty of EtCO2 as a predictor of outcomes. Importantly, since EtCO2 measurements are standard of care, this clinical variable is readily available in the perioperative period.

However, there are two major issues remaining in this study:

Due to the potential of CO2 retention in laparoscopic procedures, inclusion of these procedures next to open procedures may be problematic. The impact or lack thereof of these studies should be examined specifically i.e. using sensitivity analysis.

Response: Thank you very much for your valuable comments.

Due to the potential for CO2 retention and systemic CO2 absorption from the Trendelenburg position, anesthesiologists may likely to target higher EtCO2 values in laparoscopic surgery. In the subgroup analysis of laparoscopic versus open surgery, the point estimates of the risk ratio for organ dysfunction were exceeded than 1, but were particularly large in patients who underwent laparoscopic surgery. Our data suggest that the relationship between low EtCO2 and postoperative organ dysfunction may differ depending on whether the patient underwent laparoscopic surgery or not. 

Following your recommendation, we have revised the following information in the methods, result, and discussion section(Page 9-15) and S9 Table. 

<Methods

Finally we used the same multivariable regression models in subgroup analysis of laparoscopic versus open surgery. We calculated the crude risk ratio for postoperative organ dysfunction in each subgroup and tested the interaction between subgroups and EtCO2.

Result

In the subgroup analysis of laparoscopic versus open surgery, the point estimates of the risk ratio for organ dysfunction were exceeded than 1, but were particularly large in patients who underwent laparoscopic surgery (S9 Table). 

Since the authors hypothesize that EtCO2 is mainly an indicator of perfusion a major limitation of this study is that the relation of EtCO2 and intraoperative blood pressure measurements was not analyzed in more detail. I.E. the claim is made in the discussion that hypotension and low EtCO2, result in intraoperative hypoperfusion and postoperative organ dysfunction but only a mean MAP is shown in Table 1 without further analysis. Please include a more formal examination of EtCO2 to MAP e.g. using mean MAP and time below a threshold MAP in parallel to the model used for EtCO2. Along the same lines intraoperative vasopressor use and lactate levels should be included in the analysis.

Response: Thank you for your helpful comment. Ventilatory conditions and hemodynamic states determine EtCO2 .Hemodynamic states are determined by the preload, afterload, and cardiac output. The information about fluid volume, transfusion volume, vasopressor, and cardiac output is important. We only adjusted the MAP as the confounding factor, because we believe that the mean of MAP reflects the overall intraoperative hemodynamic situation, resulting from the overall infusion volume, transfusion volume, vasopressor, and cardiac output. 

As you highlighted, mean blood pressure may not adequately reflect intraoperative hemodynamics. In a systematic review of the relationship between intraoperative hypotension and postoperative outcomes, prolonged (>10 minutes) exposure to MAP <80 mmHg and shorter exposure to MAP <70 mmHg were associated with a mildly increased risk of end-organ injury. MAP <65–60 mmHg or 55–50 mmHg for a longer period was associated with higher injury risk[1]. In previous studies evaluating the association between blood pressure and postoperative 30-day mortality, absolute hypotension (time and cumulative duration of MAP <50 mmHg) and relative hypotension from baseline (duration of >50% reduction in MAP) were also significantly associated with increased mortality[2, 3]. 

As you have suggested, instead of the mean blood pressure, we have adjusted for relative hypotension (duration and area below threshold of blood pressure 50% below preoperative blood pressure) and absolute hypotension (duration and area below threshold of MAP below 50 mmHg) in our model. The results were the same as in the original analysis.

In our dataset of major abdominal surgery, intraoperative lactate levels were measured in 3476 of the 4171 patients. After adjusting for confounding factors by including lactate in the adjusted model, we found the same association between EtCO2 and organ dysfunction as in the original analysis.

Following your recommendation, we have revised the following information in the methods and result (Page 9-15). 

<Methods

We performed a sensitivity analysis to assess the robustness of our findings. To assess the plausibility of the primary analysis, we performed a multivariate analysis using the sensitivity model described above as model 1: (i) patients for whom arterial gases were measured during surgery (ii) patients for whom minute ventilation data were available. For patients with intraoperative arterial gas measurements, the arterial partial pressure of carbon dioxide (PaCO2)-EtCO2 gradient was added to the covariates in model 1, defined as model 5 , and the intraoperative maximum lactate value was added, defined as model 6, to investigate the relationship between intraoperative EtCO2 and postoperative organ dysfunction.

Result 

Sensitivity analysis

In the sensitivity analysis, the association between low EtCO2 and postoperative organ dysfunction was observed even when the study was restricted to patients who had arterial gas measurements during surgery and for whom minute ventilation data were available. Even after adjusting for the PaCO2-EtCO2 gradient ,intraoperative maximal lactate and minute ventilation , low EtCO2 was associated with increased postoperative organ dysfunction (S7 Table).

Minor issues:

• Please include p values for comparisons of groups in Table 1

Response:　Thank you for this comment. Following your recommendation, we have added p values for comparison groups in the Table1.

Reviewer #1

Response to Reviewer 1

We wish to express our appreciation to the reviewer for the insightful comments that have helped us improve our manuscript significantly. We agree with all the comments and have incorporated them into the revised version of our manuscript (R1).

Reviewers' comments:

Reviewer's Responses to Questions

Comments to the Author

1. Is the manuscript technically sound, and do the data support the conclusions?

Reviewer #1: Partly

2. Has the statistical analysis been performed appropriately and rigorously? 

Reviewer #1: Yes

3. Have the authors made all data underlying the findings in their manuscript fully available?

Reviewer #1: No

4. Is the manuscript presented in an intelligible fashion and written in standard English?

Reviewer #1: Yes

5. Review Comments to the Author

Reviewer #1: The authors reported about the results of cohort study based on an existing data set to investigate the association between intraoperative end-tidal carbon dioxide and postoperative organ dysfunction. They concluded that intraoperative low EtCO2 of below 35 mmHg was associated with increased postoperative organ dysfunction.

To my understanding some aspects - in particular with respect to the results should be elaborated.

First, the term "retrospective" connected to the study design should be avoided all through the paper according to the STROBE guideline for cohort studies.

Response: Thank you for this comment. We have rewritten the term "retrospective" as “historical “.

Please organize the paper according to the STROBE guideline for cohort studies and provide the checklist.

Response: Thank you for this comment. We have added the checklist.

I have some problems with understanding the "mean EtCO2" and its dichotomization to below and above 35 mmHG. Exclusion of EtCO2 < 20 mmHG (as artefact) has major impact on the number of cases with low EtCO2, as the mean EtCO2 is increased. The number of measurements also impacts the rate. As mean EtCO2 "smoothers" the effect I ask myself, if this is the clinical important effect. And an argument for the cutoff might be important. Note that dichotomization is in general problematic, for instance because of measurement errors.

Response: We excluded EtCO2 as an artifact because it would be only account for less than 1% of the EtCO2 from the start of surgery to the end of surgery and would not significantly affect the number of cases with low EtCO2. In all cases, EtCO2 were measured every 5 s.

We also performed multivariable analysis without excluding EtCO2 < 20 mmHg.The results were the same as in the original analysis.

In addition, to avoid "smoothers" the effect on mean EtCO2, we examined both time and cumulative effect of low EtCO2.

Finally, the cutoff value for this study was determined to be 35 mmHg for the following reasons:

(1) Clinically rational judgment: because the normal value of EtCO2 in the healthy patient population is about 3-5 mmHg lower than PaCO2 normal value (35-45 mmHg ), the EtCO2 normal value could be interpreted as 35 mmHg or lower. However, because our study included patients with high-risk surgery, elderly patients, and patients with pulmonary complications, setting the EtCO2 normal value above 35 mmHg by expanding the PaCO2-EtCO2 gradient would be more appropariate. 

(2) Cutoff according to a previous study: The cutoff proposed by Way and Hill is 35 mmHg[4].

(3)Confirmation by a spline curve: Our previous study examined the association between EtCO2 and mortality at 90 days postoperatively and created a spline curve between EtCO2 and mortality to confirm the validity of defining the EtCO2 cutoff level as 35 mmHg[5]. There was a linear relationship between mean EtCO2 and mortality at 90 days postoperatively, with mortality at 90 days decreasing steeply until EtCO2 reached about 35 mmHg and then flattening out. 

Similarly, the spline curve between EtCO2 and postoperative organ dysfunction confirmed the linear relationship.

Furthermore, we found a similar association between mean EtCO2 and postoperative organ dysfunction as in the original analysis.

Of course, as you indicated, further studies will be required to determine the optimal cutoff level for EtCO2.

Second in the definition of the primary endpoint variable the time to event is not reflected. This might be worth to analyze. Moreover just counting events assumes the similar importance of the events, which is rarely true in practice. So why not study the events separately, with time to event models.

A: The primary endpoint in this study was defined as postoperative organ dysfunction within one week, and we did not perform a survival analysis due to the short time period and the limitations of the database.

6. PLOS authors have the option to publish the peer review history of their article (what does this mean?). If published, this will include your full peer review and any attached files.

Do you want your identity to be public for this peer review? For information about this choice, including consent withdrawal, please see our Privacy Policy.

Reviewer #1: No

Reference:

1. Wesselink EM, Kappen TH, Torn HM, Slooter AJC, van Klei WA. Intraoperative hypotension and the risk of postoperative adverse outcomes: a systematic review. Br J Anaesth. 2018;121(4):706-21. Epub 2018/09/22. doi: 10.1016/j.bja.2018.04.036. PubMed PMID: 30236233.

2. Monk TG, Bronsert MR, Henderson WG, Mangione MP, Sum-Ping STJ, Bentt DR, et al. Association between intraoperative hypotension and hypertension and 30-day postoperative mortality in noncardiac surgery. Anesthesiology. 2015;123(2):307-19. doi: 10.1097/ALN.0000000000000756. PubMed PMID: 26083768.

3. Mascha EJ, Yang D, Weiss S, Sessler DI. Intraoperative mean arterial pressure variability and 30-day mortality in patients having noncardiac surgery. Anesthesiology. 2015;123(1):79-91. doi: 10.1097/ALN.0000000000000686. PubMed PMID: 25929547.

4. Way M, Hill GE. Intraoperative end-tidal carbon dioxide concentrations: what is the target? Anesthesiol Res Pract. 2011;2011:271539. Epub 2011/11/24. doi: 10.1155/2011/271539. PubMed PMID: 22110496; PubMed Central PMCID: PMCPMC3202118.

5. Dong L, Mizota T. In reply: Low levels of end-tidal carbon dioxide during general anesthesia and postoperative mortality. Can J Anaesth. 2022;69(3):391-2. doi: 10.1007/s12630-021-02155-3. PubMed PMID: 34817801.

---

## [Decision Letter · Decision Letter 1]

11 Nov 2022

PONE-D-22-12191R1Association between Intraoperative End-Tidal Carbon Dioxide and Postoperative Organ Dysfunction in Major Abdominal Surgery: A Historical Cohort StudyPLOS ONE

Dear Dr. Dong,

Thank you for submitting your manuscript to PLOS ONE. After careful consideration, we feel that it has merit but does not fully meet PLOS ONE’s publication criteria as it currently stands. Therefore, we invite you to submit a revised version of the manuscript that addresses the points raised during the review process. Please address the comments by reviewer #1 regarding wording, STROBE guideline adhesion and treatment of continuous variables.

We look forward to receiving your revised manuscript.

Kind regards,

Jörn Karhausen

Academic Editor

PLOS ONE

Journal Requirements:

Additional Editor Comments:

I thank the authors for the detailed responses and additional analyses following my comments. My only remaining point is the sentence (p.13) "In the subgroup analysis of laparoscopic versus open surgery, the point estimates of the risk ratio for organ dysfunction were exceeded than 1 [...]". The meaning of this sentence is not clear, please revise.

Reviewers' comments:

Reviewer's Responses to Questions

**Comments to the Author**

1. If the authors have adequately addressed your comments raised in a previous round of review and you feel that this manuscript is now acceptable for publication, you may indicate that here to bypass the “Comments to the Author” section, enter your conflict of interest statement in the “Confidential to Editor” section, and submit your "Accept" recommendation.

Reviewer #1: (No Response)

2. Is the manuscript technically sound, and do the data support the conclusions?

Reviewer #1: No

3. Has the statistical analysis been performed appropriately and rigorously? 

Reviewer #1: Yes

4. Have the authors made all data underlying the findings in their manuscript fully available?

Reviewer #1: No

5. Is the manuscript presented in an intelligible fashion and written in standard English?

Reviewer #1: No

6. Review Comments to the Author

Reviewer #1: Strobe Checklist is not appended and the paper is not structured accordingly.

Historical cohort study is not a scientific term to describe a study design. So as suggested, that study appears to be a consort study based on a medical records or registry. So the title " Historical cohort" is not specific.

Categorization of continuous variables introduces uncertainty in the definition of the primary endpoint variable. So this needs scientific justification, e.g. by refering to a regulatory guideline or consensus of medical discipline.

7. PLOS authors have the option to publish the peer review history of their article (what does this mean?). If published, this will include your full peer review and any attached files.

Reviewer #1: No

---

## [Author Response · Author response to Decision Letter 1]

28 Dec 2022

Academic Editor

PLOS ONE

Dear Prof. Jörn Karhausen:

We would like to thank all of you for providing us with valuable comments on our manuscript entitled “Association between Intraoperative End-Tidal Carbon Dioxide and Postoperative Organ Dysfunction in Major Abdominal Surgery: A Cohort Study” We have responded to all the reviewers' comments. Our itemized responses are listed below. We hope this revision process has remedied all the concerns related to the previous version of the manuscript. Changes or corrections of the original manuscript have been highlighted in red in the revised manuscript. The revised manuscript has also been checked and revised by a native English-speaking medical-editing professional.

We hope your response to our revised manuscript will be positive. Thank you once again for reviewing it.

Sincerely,

Correspondence Author: 

Li Dong, M.D.Ph.D.

Department of Anaesthesia, Kyoto University Hospital, 54 Shogoin-Kawahara-cho, Kyoto 606-8507, JAPAN

Tel: +81-75-751-3433

E-mail: dongli@kuhp.kyoto-u.ac.jp

Additional Editor Comments:

I thank the authors for the detailed responses and additional analyses following my comments. My only remaining point is the sentence (p.13) "In the subgroup analysis of laparoscopic versus open surgery, the point estimates of the risk ratio for organ dysfunction were exceeded than 1 but were particularly large in patients who underwent laparoscopic surgery (S9 Table)". The meaning of this sentence is not clear, please revise.

Response: Thank you for this comment. We have rewritten the sentence(p.13) as 

In the subgroup analysis of laparoscopic versus open surgery, the point estimates of the risk ratio for organ dysfunction were exceeded than 1 (S9 Table). 

Response to Additional Editor 

We wish to express our appreciation to the referee for the insightful comments that have helped us improve our manuscript significantly. We agree with all the comments and have incorporated them into the revised version of our manuscript (R2).

Reviewers' comments:

Reviewer's Responses to Questions

Comments to the Author

1. If the authors have adequately addressed your comments raised in a previous round of review and you feel that this manuscript is now acceptable for publication, you may indicate that here to bypass the “Comments to the Author” section, enter your conflict of interest statement in the “Confidential to Editor” section, and submit your "Accept" recommendation.

Reviewer #1: (No Response)

2. Is the manuscript technically sound, and do the data support the conclusions?

Reviewer #1: No

3. Has the statistical analysis been performed appropriately and rigorously? 

Reviewer #1: Yes

4. Have the authors made all data underlying the findings in their manuscript fully available?

Reviewer #1: No

5. Is the manuscript presented in an intelligible fashion and written in standard English?

Reviewer #1: No

6. Review Comments to the Author

Reviewer #1: Strobe Checklist is not appended and the paper is not structured accordingly.

Response: Thank you for this comment. We have added the STROBE checklist.

Historical cohort study is not a scientific term to describe a study design. So as suggested, that study appears to be a consort study based on a medical records or registry. So the title " Historical cohort" is not specific.

Response: Thank you for this comment. In compliance with STROBE [1], we refrained from using the term retrospective or historical and rewrote it as cohort study. We have rewritten the term " historical cohort study " as "cohort study".

Categorization of continuous variables introduces uncertainty in the definition of the primary endpoint variable. So this needs scientific justification, e.g. by refering to a regulatory guideline or consensus of medical discipline.

Response: Thank you for this comment. The cutoff value for this study was determined to be 35 mmHg referring to consensus of medical discipline.:

(1) Clinically rational judgment: Hypocapnia, also known as hypocarbia, is a decrease in blood carbon dioxide (PaCO2) concentration below 35 mmHg, the normal reference range, as described in textbook.

(https://www.ncbi.nlm.nih.gov/books/NBK493167/#_NBK493167_pubdet_). 

Furthermore, the normal value of end-tidal CO2 (EtCO2), the partial pressure of carbon dioxide at the end of expiration detected by capnometry, is usually 35-45 mmHg.

(https://www.ems1.com/ems-products/capnography/articles/5-things-to-know-about-capnography-Hr5ETRdXzCoU3fLH/)

(2) Cutoff according to a previous study: In a systematic review of parallel-arm randomized controlled trials(RCT) comparing hypercapnia and normocapnia in adult patients undergoing general anesthesia, 7 of 10 RCT definitions of cutoff were set at PaCO2 or EtCO2 as 35 mmHg, while other cutoffs were defined as EtCO2 33 mmHg,37 mmHg and PaCO2 36-40 mmHg[2]. 

Following your recommendation, we have revised the following information in the methods(Page6).

<Method

Dose effects were assessed using the mean EtCO2; patients were divided into two groups based on the cutoff level of 35 mmHg , widely used lower limit of normal PaCO2[2] [3].>

7. PLOS authors have the option to publish the peer review history of their article (what does this mean?). If published, this will include your full peer review and any attached files.

Do you want your identity to be public for this peer review? For information about this choice, including consent withdrawal, please see our Privacy Policy.

Reviewer #1: No

1. Vandenbroucke JP, von Elm E, Altman DG, Gøtzsche PC, Mulrow CD, Pocock SJ, et al. Strengthening the Reporting of Observational Studies in Epidemiology (STROBE): explanation and elaboration. Int J Surg. 2014;12(12):1500-24. Epub 2014/07/22. doi: 10.1016/j.ijsu.2014.07.014. PubMed PMID: 25046751.

2. Petran J, Ansems K, Rossaint R, Marx G, Kalvelage C, Kopp R, et al. Effects of hypercapnia versus normocapnia during general anesthesia on outcomes: a systematic review and meta-analysis. Brazilian Journal of Anesthesiology (English Edition). 2022;72(3):398-406. doi: https://doi.org/10.1016/j.bjane.2020.11.010.

3. Way M, Hill GE. Intraoperative end-tidal carbon dioxide concentrations: what is the target? Anesthesiol Res Pract. 2011;2011:271539. Epub 2011/11/24. doi: 10.1155/2011/271539. PubMed PMID: 22110496; PubMed Central PMCID: PMCPMC3202118.

---

## [Editor Report · Decision Letter 2]

4 Jan 2023

PONE-D-22-12191R2

Association between Intraoperative End-Tidal Carbon Dioxide and Postoperative Organ Dysfunction in Major Abdominal Surgery: A Cohort Study

PLOS ONE

Dear Dr. Dong,

Thank you for submitting your manuscript to PLOS ONE. After careful consideration, we feel that it has merit but does not fully meet PLOS ONE’s publication criteria as it currently stands. Therefore, we invite you to submit a revised version of the manuscript that addresses the points raised during the review process.

We look forward to receiving your revised manuscript.

Kind regards,

Jörn Karhausen

Academic Editor

PLOS ONE

Journal Requirements:

Additional Editor Comments:

"In the subgroup analysis of laparoscopic versus open surgery, the point estimates of the risk ratio for organ dysfunction were exceeded than 1 (S9 Table)." Sorry to insist, but the sentence is still not clear. Do you mean ...the risk ratio exceeded 1?

Otherwise, all comments appear to be fully addressed. Thank you!

---

## [Author Response · Author response to Decision Letter 2]

19 Jan 2023

Additional Editor Comments:

"In the subgroup analysis of laparoscopic versus open surgery, the point estimates of the risk ratio for organ dysfunction were exceeded than 1 (S9 Table)." Sorry to insist, but the sentence is still not clear. Do you mean ...the risk ratio exceeded 1?

Otherwise, all comments appear to be fully addressed. Thank you!

Response: Thank you very much for your valuable comments.

In a subgroup analysis of laparoscopic versus open surgery, the adjusted risk ratio for laparoscopic surgery was 1.42 (adjusted risk ratio, 1.42; 95% confidence interval [CI], 1.12-1.80; p = 0.004), and for open surgery adjusted risk ratio, 1.14; 95% confidence interval [CI], 0.91-1.42; p = 0.237). Although the small sample size in the subgroup analysis makes it difficult to find significant differences, the point estimate of the risk ratio for organ dysfunction was exceeded than 1.

---

## [Editor Report · Decision Letter 3]

2 Feb 2023

Association between Intraoperative End-Tidal Carbon Dioxide and Postoperative Organ Dysfunction in Major Abdominal Surgery: A Cohort Study

PONE-D-22-12191R3

Dear Dr. Dong,

We’re pleased to inform you that your manuscript has been judged scientifically suitable for publication and will be formally accepted for publication once it meets all outstanding technical requirements.

Kind regards,

Jörn Karhausen

Academic Editor

PLOS ONE
---

## [Editor Report · Acceptance letter]

1 Mar 2023

PONE-D-22-12191R3 

Association between Intraoperative End-Tidal Carbon Dioxide and Postoperative Organ Dysfunction in Major Abdominal Surgery: A Cohort Study 

Dear Dr. Dong:

I'm pleased to inform you that your manuscript has been deemed suitable for publication in PLOS ONE. Congratulations! Your manuscript is now with our production department. 

Kind regards, 

on behalf of

Dr. Jörn Karhausen 

Academic Editor

PLOS ONE